# Heterogeneous-Neighborhood-based Multi-Task Local Learning Algorithms

**Yu Zhang**
Department of Computer Science, Hong Kong Baptist University
`yuzhang@comp.hkbu.edu.hk`

## Abstract

All the existing multi-task local learning methods are defined on homogeneous neighborhood which consists of all data points from only one task. In this paper, different from existing methods, we propose local learning methods for multi-task classification and regression problems based on heterogeneous neighborhood which is defined on data points from all tasks. Specifically, we extend the $k$-nearest-neighbor classifier by formulating the decision function for each data point as a weighted voting among the neighbors from all tasks where the weights are task-specific. By defining a regularizer to enforce the task-specific weight matrix to approach a symmetric one, a regularized objective function is proposed and an efficient coordinate descent method is developed to solve it. For regression problems, we extend the kernel regression to multi-task setting in a similar way to the classification case. Experiments on some toy data and real-world datasets demonstrate the effectiveness of our proposed methods.

## 1   Introduction

For single-task learning, besides global learning methods there are local learning methods [7], e.g., $k$-nearest-neighbor (KNN) classifier and kernel regression. Different from the global learning methods, the local learning methods make use of locality structure in different regions of the feature space and are complementary to the global learning algorithms. In many applications, the single-task local learning methods have shown comparable performance with the global counterparts. Moreover, besides classification and regression problems, the local learning methods are also applied to some other learning problems, e.g., clustering [18] and dimensionality reduction [19]. When the number of labeled data is not very large, the performance of the local learning methods is limited due to sparse local density [14]. In this case, we can leverage the useful information from other related tasks to help improve the performance which matches the philosophy of multi-task learning [8, 4, 16]. Multi-task learning utilizes supervised information from some related tasks to improve the performance of one task at hand and during the past decades many advanced methods have been proposed for multi-task learning, e.g., [17, 3, 9, 1, 2, 6, 12, 20, 14, 13]. Among those methods, [17, 14] are two representative multi-task local learning methods. Even though both methods in [17, 14] use KNN as the base learner for each task, Thrun and O'Sullivan [17] focus on learning cluster structure among different tasks while Parameswaran and Weinberger [14] learn different distance metrics for different tasks. The KNN classifiers use in both two methods are defined on the homogeneous neighborhood which is the set of nearest data points from the same task the query point belongs to. In some situation, it is better to use a heterogeneous neighborhood which is defined as the set of nearest data points from *all* tasks. For example, suppose we have two similar tasks marked with two colors as shown in Figure 1. For a test data point marked with '?' from one task, we obtain an estimation with low confidence or even a wrong one based on the homogeneous neighborhood. However, if we can use the data points from both two tasks to define the neighborhood (i.e., heterogeneous neighborhood), we can obtain a more confident estimation.

In this paper, we propose novel local learning models for multi-task learning based on the heterogeneous neighborhood. For multi-task classification problems, we extend the KNN classifier by formulating the decision function on each data point as weighted voting of its neighbors from all tasks where the weights are task-specific. Since multi-task learning usually considers that the contribution of one task to another one equals that in the reverse direction, we define a regularizer to enforce the task-specific weight matrix to approach a symmetric matrix and then based on this regularizer, a regularized objective function is proposed. We develop an efficient coordinate descent method to solve it. Moreover, we also propose a local method for multi-task regression problems. Specifically, we extend the kernel regression method to multi-task setting in a similar way to the classification case. Experiments on some toy data and real-world datasets demonstrate the effectiveness of our proposed methods.

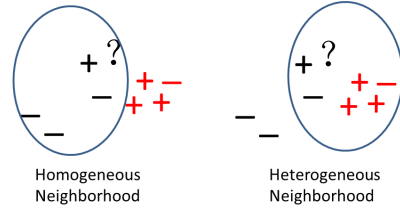

Figure 1: Data points with one color (i.e., black or red) are from the same task and those with one type of marker (i.e., '+' or '-') are from the same class. A test data point is represented by '?'.

## 2   A Multi-Task Local Classifier based on Heterogeneous Neighborhood

In this section, we propose a local classifier for multi-task learning by generalizing the KNN algorithm, which is one of the most widely used local classifiers for single-task learning.

Suppose we are given $m$ learning tasks $\{\mathcal{T}_i\}_{i=1}^m$. The training set consists of $n$ triples $(\mathbf{x}_i, y_i, t_i)$ with the $i$th data point as $\mathbf{x}_i \in \mathbb{R}^D$, its label $y_i \in \{-1, 1\}$ and its task indicator $t_i \in \{1, \ldots, m\}$. So each task is a binary classification problem with $n_i = |\{j | t_j = i\}|$ data points belonging to the $i$th task $\mathcal{T}_i$.

For the $i$th data point $\mathbf{x}_i$, we use $\mathcal{N}_k(i)$ to denote the set of the indices of its $k$ nearest neighbors. If $\mathcal{N}_k(i)$ is a homogeneous neighborhood which only contains data points from the task that $\mathbf{x}_i$ belongs to, we can use $d(\mathbf{x}_i) = \text{sgn}\left(\sum_{j \in \mathcal{N}_k(i)} s(i, j) y_j\right)$ to make a decision for $\mathbf{x}_i$ where $\text{sgn}(\cdot)$ denotes the sign function and $s(i, j)$ denotes a similarity function between $\mathbf{x}_i$ and $\mathbf{x}_j$. Here, by defining $\mathcal{N}_k(i)$ as a heterogeneous neighborhood which contains data points from all tasks, we cannot directly utilize this decision function and instead we introduce a weighted decision function by using task-specific weights as

$$d(\mathbf{x}_i) = \text{sgn}\left(\sum_{j \in \mathcal{N}_k(i)} w_{t_i, t_j} s(i, j) y_j\right)$$

where $w_{qr}$ represents the contribution of the $r$th task $\mathcal{T}_r$ to the $q$th one $\mathcal{T}_q$ when $\mathcal{T}_r$ has some data points to be neighbors of a data point from $\mathcal{T}_q$. Of course, the contribution from one task to itself should be positive and also the largest, i.e., $w_{ii} \geq 0$ and $-w_{ii} \leq w_{ij} \leq w_{ii}$ for $j \neq i$. When $w_{qr}(q \neq r)$ approaches $w_{qq}$, it means $\mathcal{T}_r$ is very similar to $\mathcal{T}_q$ in local regions. At another extreme where $w_{qr}(q \neq r)$ approaches $-w_{qq}$, if we flip the labels of data points in $\mathcal{T}_r$, $\mathcal{T}_r$ can have a positive contribution $-w_{qr}$ to $\mathcal{T}_q$ which indicates that $\mathcal{T}_r$ is negatively correlated to $\mathcal{T}_q$. Moreover, when $w_{qr}/w_{qq}(q \neq r)$ is close to 0 which implies there is no contribution from $\mathcal{T}_r$ to $\mathcal{T}_q$, $\mathcal{T}_r$ is likely to be unrelated to $\mathcal{T}_q$. So the utilization of $\{w_{qr}\}$ can model three task relationships: positive task correlation, negative task correlation and task unrelatedness as in [6, 20].

We use $f(\mathbf{x}_i)$ to define the estimation function as $f(\mathbf{x}_i) = \sum_{j \in \mathcal{N}_k(i)} w_{t_i, t_j} s(i, j) y_j$. Then similar to support vector machine (SVM), we use hinge loss $l(y, y') = \max(0, 1 - yy')$ to measure empirical performance on the training data. Moreover, recall that $w_{qr}$ represents the contribution of $\mathcal{T}_r$ to $\mathcal{T}_q$ and $w_{rq}$ is the contribution of $\mathcal{T}_q$ to $\mathcal{T}_r$. Since multi-task learning usually considers that the contribution of $\mathcal{T}_r$ to $\mathcal{T}_q$ almost equals that of $\mathcal{T}_q$ to $\mathcal{T}_r$, we expect $w_{qr}$ to be close to $w_{rq}$. To encode this priori information into our model, we can either formulate it as $w_{qr} = w_{rq}$, a hard constraint, or a soft regularizer, i.e., minimizing $(w_{qr} - w_{rq})^2$ to enforce $w_{qr} \approx w_{rq}$, which is more preferred. Combining all the above considerations, we can construct a objective function for our proposed method MT-KNN as

$$\min_{\mathbf{W}} \sum_{i=1}^n l(y_i, f(\mathbf{x}_i)) + \frac{\lambda_1}{4} \|\mathbf{W} - \mathbf{W}^T\|_F^2 + \frac{\lambda_2}{2} \|\mathbf{W}\|_F^2 \quad \text{s.t. } w_{qq} \geq 0, \ w_{qq} \geq w_{qr} \geq -w_{qq} \ (q \neq r) \quad (1)$$

where $\mathbf{W}$ is a $m \times m$ matrix with $w_{qr}$ as its $(q,r)$th element and $\|\cdot\|_F$ denotes Frobenius norm of a matrix. The first term in the objective function of problem (1) measures the training loss, the second one enforces $\mathbf{W}$ to be a symmetric matrix which implies $w_{qr} \approx w_{rq}$, and the last one penalizes the complexity of $\mathbf{W}$. The regularization parameters $\lambda_1$ and $\lambda_2$ balance the trade-off between these three terms.

## 2.1 Optimization Procedure

In this section, we discuss how to solve problem (1). We first rewrite $f(\mathbf{x}_i)$ as $f(\mathbf{x}_i) = \sum_{j=1}^{m} w_{t_i j} \left( \sum_{l \in \mathcal{N}_k^j(i)} s(i,l) y_l \right) = \mathbf{w}_{t_i} \hat{\mathbf{x}}_i$ where $\mathcal{N}_k^j(i)$ denotes the set of the indices of $\mathbf{x}_i$'s nearest neighbors from the $j$th task in $\mathcal{N}_k(i)$, $\mathbf{w}_{t_i} = (w_{t_i 1}, \ldots, w_{t_i m})$ is the $t_i$th row of $\mathbf{W}$, and $\hat{\mathbf{x}}_i$ is a $m \times 1$ vector with the $j$th element as $\sum_{l \in \mathcal{N}_k^j(i)} s(i,l) y_l$. Then we can reformulate problem (1) as

$$\min_{\mathbf{W}} \sum_{i=1}^{m} \sum_{j \in \mathcal{T}_i} l(y_j, \mathbf{w}_i \hat{\mathbf{x}}_j) + \frac{\lambda_1}{4} \|\mathbf{W} - \mathbf{W}^T\|_F^2 + \frac{\lambda_2}{2} \|\mathbf{W}\|_F^2 \quad \text{s.t. } w_{qq} \geq 0, \ w_{qq} \geq w_{qr} \geq -w_{qq}(q \neq r).$$

(2)

To solve problem (2), we use a coordinate descent method, which is also named as an alternating optimization method in some literatures.

By adopting the hinge loss in problem (2), the optimization problem for $w_{ik}$ $(k \neq i)$ is formulated as

$$\min_{w_{ik}} \frac{\lambda}{2} w_{ik}^2 - \beta_{ik} w_{ik} + \sum_{j \in \mathcal{T}_i} \max(0, a_{ik}^j w_{ik} + b_{ik}^j) \quad \text{s.t. } c_{ik} \leq w_{ik} \leq e_{ik}$$

(3)

where $\lambda = \lambda_1 + \lambda_2$, $\beta_{ik} = \lambda_1 w_{ki}$, $\hat{x}_{jk}$ is the $k$th element of $\hat{\mathbf{x}}_j$, $a_{ik}^j = -y_j \hat{x}_{jk}$, $b_{ik}^j = 1 - y_j \sum_{t \neq k} w_{it} \hat{x}_{jt}$, $c_{ik} = -w_{ii}$, and $e_{ik} = w_{ii}$. If the objective function of problem (3) only has the first two terms, this problem will become a univariate quadratic programming (QP) problem with a linear inequality constraint, leading to an analytical solution. Moreover, similar to SVM we can introduce some slack variables for the third term in the objective function of problem (3) and then that problem will become a QP problem with $n_i + 1$ variables and $2n_i + 1$ linear constraints. We can use off-the-shelf softwares to solve this problem in polynomial time. However, the whole optimization procedure may not be very efficient since we need to solve problem (3) and call QP solvers for multiple times. Here we utilize the piecewise linear structure of the last term in the objective function of problem (3) and propose a more efficient solution.

We assume all $a^j$ are non-zero and otherwise we can discard them without affecting the solution since the corresponding losses are constants. We define six index sets as

$$\mathcal{C}_1 = \{j | a_{ik}^j > 0, -\frac{b_{ik}^j}{a_{ik}^j} < c_{ik}\}, \ \mathcal{C}_2 = \{j | a_{ik}^j > 0, c_{ik} \leq -\frac{b_{ik}^j}{a_{ik}^j} \leq e_{ik}\}, \ \mathcal{C}_3 = \{j | a_{ik}^j > 0, -\frac{b_{ik}^j}{a_{ik}^j} > e_{ik}\}$$

$$\mathcal{C}_4 = \{j | a_{ik}^j < 0, -\frac{b_{ik}^j}{a_{ik}^j} < c_{ik}\}, \ \mathcal{C}_5 = \{j | a_{ik}^j < 0, c_{ik} \leq -\frac{b_{ik}^j}{a_{ik}^j} \leq e_{ik}\}, \ \mathcal{C}_6 = \{j | a_{ik}^j < 0, -\frac{b_{ik}^j}{a_{ik}^j} > e_{ik}\}.$$

It is easy to show that when $j \in \mathcal{C}_1 \cup \mathcal{C}_6$ where the operator $\cup$ denotes the union of sets, $a_{ik}^j w + b_{ik}^j > 0$ holds for $w \in [c_{ik}, e_{ik}]$, corresponding to the set of data points with non-zero loss. Oppositely when $j \in \mathcal{C}_3 \cup \mathcal{C}_4$, the values of the corresponding losses become zero since $a_{ik}^j w + b_{ik}^j \leq 0$ holds for $w \in [c_{ik}, e_{ik}]$. The variation lies in the data points with indices $j \in \mathcal{C}_2 \cup \mathcal{C}_5$. We sort sequence $\{-b_{ik}^j/a_{ik}^j | j \in \mathcal{C}_2\}$ and record it in a vector $\mathbf{u}$ of length $d_u$ with $u_1 \leq \ldots \leq u_{d_u}$. Moreover, we also keep a index mapping $\mathbf{q}_u$ with its $r$th element $q_r^u$ defined as $q_r^u = j$ if $u_r = -b_{ik}^j/a_{ik}^j$. Similarly, for sequence $\{-b_{ik}^j/a_{ik}^j | j \in \mathcal{C}_5\}$, we define a sorted vector $\mathbf{v}$ of length $d_v$ and the corresponding index mapping $\mathbf{q}_v$. By using the merge-sort algorithm, we merge $\mathbf{u}$ and $\mathbf{v}$ into a sorted vector $\mathbf{s}$ and then we add $c_{ik}$ and $e_{ik}$ into $\mathbf{s}$ as the minimum and maximum elements if they are not contained in $\mathbf{s}$. Obviously, in range $[s_l, s_{l+1}]$ where $s_l$ is the $l$th element of $\mathbf{s}$ and $d_s$ is the length of $\mathbf{s}$, problem (3) becomes a univariate QP problem which has an analytical solution. So we can compute local minimums in successive regions $[s_l, s_{l+1}]$ $(l = 1, \ldots, d_s - 1)$ and get the global minimum over region $[c_{ik}, e_{ik}]$ by comparing all local optima. The key operation is to compute the coefficients of quadratic function over each region $[s_l, s_{l+1}]$ and we devise an algorithm in Table 1 which only needs to scan $\mathbf{s}$ once, leading to an efficient solution for problem (3).

The first step of the algorithm in Table 1 needs $O(n_i)$ time complexity to construct the six sets $\mathcal{C}_1$ to $\mathcal{C}_6$. In step 2, we need to sort two sequences to obtain $\mathbf{u}$ and $\mathbf{v}$ in $O(d_u \ln d_u + d_v \ln d_v)$ time and merge two sequences to get $\mathbf{s}$ in $O(d_u + d_v)$. Then it costs $O(n_i)$ to calculate coefficients $c_0$ and $c_1$ by scanning $\mathcal{C}_1$, $\mathcal{C}_2$ and $\mathcal{C}_6$ in step 4 and 5. Then from step 6 to step 13, we need to scan vector $\mathbf{s}$ once which costs $O(d_u + d_v)$ time. The overall complexity of the algorithm in Table 1 is $O(d_u \ln d_u + d_v \ln d_v + n_i)$ which is at most $O(n_i \ln n_i)$ due to $d_u + d_v \leq n_i$.

For $w_{ii}$, the optimization problem is formulated as

$$\min_{w_{ii}} \frac{\lambda_2}{2} w_{ii}^2 + \sum_{j \in \mathcal{T}_i} \max(0, a_i^j w_{ii} + b_i^j) \quad \text{s.t. } w_{ii} \geq c_i, \quad (4)$$

where $a_i^j = -y_j \hat{x}_{ji}$, $b_i^j = 1 - y_j \sum_{t \neq i} w_{it} \hat{x}_{jt}$, $c_i = \max(0, \max_{j \neq i}(|w_{ij}|))$, and $|\cdot|$ denotes the absolute value of a scalar. The main difference between problem (3)

Table 1: Algorithm for problem (3)

01: Construct four sets $\mathcal{C}_1, \mathcal{C}_2, \mathcal{C}_3, \mathcal{C}_4, \mathcal{C}_5$ and $\mathcal{C}_6$;
02: Construct $\mathbf{u}, \mathbf{q}_u, \mathbf{v}, \mathbf{q}_v$ and $\mathbf{s}$;
03: Insert $c_{ik}$ and $e_{ik}$ into $\mathbf{s}$ if needed;
04: $c_0 := \sum_{j \in \mathcal{C}_1 \cup \mathcal{C}_2 \cup \mathcal{C}_6} b_{ik}^j$;
05: $c_1 := \sum_{j \in \mathcal{C}_1 \cup \mathcal{C}_2 \cup \mathcal{C}_6} a_{ik}^j - \beta_{ik}$;
06: $w := s_{d_s}$;
07: $o := c_0 + c_1 w + \lambda w^2/2$;
    for $l = d_s - 1$ to $1$
      if $s_{l+1} = u_r$ for some $r$
08:     $c_0 := c_0 - b_{ik}^{q_r^u}$; $c_1 := c_1 - a_{ik}^{q_r^u}$;
      end if
      if $s_{l+1} = v_r$ for some $r$
09:     $c_0 := c_0 + b_{ik}^{q_r^v}$; $c_1 := c_1 + a_{ik}^{q_r^v}$;
      end if
10:     $w_0 := \min(s_{l+1}, \max(s_l, -\frac{c_1}{\lambda}))$;
11:     $o_0 := c_0 + c_1 w_0 + \lambda w_0^2/2$;
      if $o_0 < o$
12:     $w := w_0$; $o := o_0$;
      end if
13:     $l := l - 1$;
    end for

and problem (4) is that there exist a box constraint for $w_{ik}$ in problem (3) but in problem (4) $w_{ii}$ is only lower-bounded. We define $e_i$ as $e_i = \max_j\{-\frac{b_i^j}{a_i^j}\}$ for all $a_i^j \neq 0$. For $w_{ii} \in [e_i, +\infty)$, the objective function of problem (4) can be reformulated as $\frac{\lambda_2}{2} w_{ii}^2 + \sum_{j \in \mathcal{S}} (a_i^j w_{ii} + b_i^j)$ where $\mathcal{S} = \{j | a_i^j > 0\}$ and the minimum value in $[e_i, +\infty)$ will take at $w_{ii}^{(1)} = \max\{e_i, -\frac{\sum_{j \in \mathcal{S}} a_i^j}{\lambda_2}\}$. Then we can use the algorithm in Table 1 to find the minimizor $w_{ii}^{(2)}$ in the interval $[c_i, e_i]$ for problem (4). Finally we can choose the optimal solution to problem (4) from $\{w_{ii}^{(1)}, w_{ii}^{(2)}\}$ by comparing the corresponding values of the objective function.

Since the complexity to solve both problem (3) and (4) is $O(n_i \ln n_i)$, the complexity of one update for the whole matrix $\mathbf{W}$ is $O(m \sum_{i=1}^{m} n_i \ln n_i)$. Usually the coordinate descent algorithm converges very fast in a small number of iterations and hence the whole algorithm to solve problem (2) or (1) is very efficient.

We can use other loss functions for problem (2) instead of hinge loss, e.g., square loss $l(s, t) = (s - t)^2$ as in the least square SVM [10]. It is easy to show that problem (3) has an analytical solution as $w_{ik} = \min\left(\max\left(c_{ik}, \frac{\beta_{ik} - 2 \sum_{j \in \mathcal{T}_i} a_{ik}^j b_{ik}^j}{\lambda + 2 \sum_{j \in \mathcal{T}_i} (a_{ik}^j)^2}\right), e_{ik}\right)$ and the solution to problem (4) can be computed as $w_{ii} = \max\left(c_i, \frac{-2 \sum_{j \in \mathcal{T}_i} a_i^j b_i^j}{\lambda_2 + 2 \sum_{j \in \mathcal{T}_i} (a_i^j)^2}\right)$. Then the computational complexity of the whole algorithm to solve problem (2) by adopting square loss is $O(mn)$.

## 3 A Multi-Task Local Regressor based on Heterogeneous Neighborhood

In this section, we consider the situation that each task is a regression problem with each label $y_i \in \mathbb{R}$.

Similar to the classification case in the previous section, one candidate for multi-task local regressor is a generalization of kernel regression, a counterpart of KNN classifier for regression problems, and the estimation function can be formulated as

$$f(\mathbf{x}_i) = \frac{\sum_{j \in \mathcal{N}_k(i)} w_{t_i, t_j} s(i, j) y_j}{\sum_{j \in \mathcal{N}_k(i)} w_{t_i, t_j} s(i, j)} \quad (5)$$

where $w_{qr}$ also represents the contribution of $\mathcal{T}_r$ to $\mathcal{T}_q$. Since the denominator of $f(\mathbf{x}_i)$ is a linear combination of elements in each row of $\mathbf{W}$ with data-dependent combination coefficients, if we utilize a similar formulation to problem (1) with square loss, we need to solve a complex and non-convex fractional programming problem. For computational consideration, we resort to another way to construct the multi-task local regressor.

Recall that the estimation function for the classification case is formulated as $f(\mathbf{x}_i) = \sum_{j=1}^{m} w_{t_ij} \left( \sum_{l \in \mathcal{N}_k^j(i)} s(i,l) y_l \right)$. We can see that the expression in the brackets on the right-hand side can be viewed as a prediction for $\mathbf{x}_i$ based on its neighbors in the $j$th task. Inspired by this observation, we can construct a prediction $\hat{y}_j^i$ for $\mathbf{x}_i$ based on its neighbors from the $j$th task by utilizing any regressor, e.g., kernel regression and support vector regression. Here due to the local nature of our proposed method, we choose the kernel regression method, which is a local regression method, as a good candidate and hence $\hat{y}_j^i$ is formulated as $\hat{y}_j^i = \frac{\sum_{l \in \mathcal{N}_k^j(i)} s(i,l) y_l}{\sum_{l \in \mathcal{N}_k^j(i)} s(i,l)}$. When $j$ equals $t_i$ which means we use neighbored data points from the task that $\mathbf{x}_i$ belongs to, we can use this prediction in confidence. However, if $j \neq t_i$, we cannot totally trust the prediction and need to add some weight $w_{t_i,j}$ as a confidence. Then by using the square loss, we formulate an optimization problem to get the estimation function $f(\mathbf{x}_i)$ based on $\{\hat{y}_j^i\}$ as

$$f(\mathbf{x}_i) = \arg \min_y \sum_{j=1}^{m} w_{t_i,j}(y - \hat{y}_j^i)^2 = \frac{\sum_{j=1}^{m} w_{t_i,j} \hat{y}_j^i}{\sum_{j=1}^{m} w_{t_i,j}}. \tag{6}$$

Compared with the regression function of the direct extension of kernel regression to multi-task learning in Eq. (5), the denominator of our proposed regressor in Eq. (6) only includes the row summation of $\mathbf{W}$, making the optimization problem easier to solve as we will see later. Since the scale of $w_{ij}$ does not matter the value of the estimation function in Eq. (6), we constrain the row summation of $\mathbf{W}$ to be 1, i.e., $\sum_{j=1}^{m} w_{ij} = 1$ for $i = 1, \ldots, m$. Moreover, the estimation $\hat{y}_{t_i}^i$ by using data from the same task as $\mathbf{x}_i$ is more trustful than the estimations based on other tasks, which suggests $w_{ii}$ should be the largest among elements in the $i$th row. Then this constraint implies that $w_{ii} \geq \frac{1}{m} \sum_k w_{ik} = \frac{1}{m} > 0$. To capture the negative task correlations, $w_{ij}$ $(i \neq j)$ is only required to be a real scalar and $w_{ij} \geq -w_{ii}$. Combining the above consideration, we formulate an optimization problem as

$$\min_{\mathbf{W}} \sum_{i=1}^{m} \sum_{j \in \mathcal{T}_i} (\mathbf{w}_i \hat{\mathbf{y}}_j - y_j)^2 + \frac{\lambda_1}{4} \|\mathbf{W} - \mathbf{W}^T\|_F^2 + \frac{\lambda_2}{2} \|\mathbf{W}\|_F^2 \text{ s.t. } \mathbf{W}\mathbf{1} = \mathbf{1}, \ w_{ii} \geq w_{ij} \geq -w_{ii}, \tag{7}$$

where $\mathbf{1}$ denotes a vector of all ones with appropriate size and $\hat{\mathbf{y}}_j = (\hat{y}_1^j, \ldots, \hat{y}_m^j)^T$. In the following section, we discuss how to optimize problem (7).

### 3.1 Optimization Procedure

Due to the linear equality constraints in problem (7), we cannot apply a coordinate descent method to update variables one by one in a similar way to problem (2). However, similar to the SMO algorithm [15] for SVM, we can update two variables in one row of $\mathbf{W}$ at one time to keep the linear equality constraints valid.

We update each row one by one and the optimization problem with respect to $\mathbf{w}_i$ is formulated as

$$\min_{\mathbf{w}_i} \frac{1}{2} \mathbf{w}_i \mathbf{A} \mathbf{w}_i^T + \mathbf{w}_i \mathbf{b}^T \quad \text{s.t. } \sum_{j=1}^{m} w_{ij} = 1, \ -w_{ii} \leq w_{ij} \leq w_{ii} \ \forall j \neq i, \tag{8}$$

where $\mathbf{A} = 2 \sum_{j \in \mathcal{T}_i} \hat{\mathbf{y}}_j \hat{\mathbf{y}}_j^T + \lambda_1 \mathbf{I}_m^i + \lambda_2 \mathbf{I}_m$, $\mathbf{I}_m$ is an $m \times m$ identity matrix, $\mathbf{I}_m^i$ is a copy of $\mathbf{I}_m$ by setting the $(i,i)$th element to be 0, $\mathbf{b} = -2 \sum_{j \in \mathcal{T}_i} y_j \hat{\mathbf{y}}_j^T - \lambda_1 \mathbf{c}_i^T$, and $\mathbf{c}_i$ is the $i$th column of $\mathbf{W}$ by setting its $i$th element to 0. We define the Lagrangian as

$$J = \frac{1}{2} \mathbf{w}_i \mathbf{A} \mathbf{w}_i^T + \mathbf{w}_i \mathbf{b}^T - \alpha(\sum_{j=1}^{m} w_{ij} - 1) - \sum_{j \neq i} (w_{ii} - w_{ij})\beta_j - \sum_{j \neq i} (w_{ii} + w_{ij})\gamma_j.$$

The Karush-Kuhn-Tucker (KKT) optimality condition is formulated as

$$\frac{\partial J}{\partial w_{ij}} = \mathbf{w}_i \mathbf{a}_j + b_j - \alpha + \beta_j - \gamma_j = 0, \text{ for } j \neq i \tag{9}$$

$$\frac{\partial J}{\partial w_{ii}} = \mathbf{w}_i \mathbf{a}_i + b_i - \alpha - \sum_{k \neq i} (\beta_k + \gamma_k) = 0 \tag{10}$$

$$\beta_j \geq 0, \ (w_{ii} - w_{ij})\beta_j = 0 \ \forall j \neq i \tag{11}$$

$$\gamma_j \geq 0, (w_{ii} + w_{ij})\gamma_j = 0 \ \forall j \neq i, \tag{12}$$

where $\mathbf{a}_j$ is the $j$th column of $\mathbf{A}$ and $b_j$ is the $j$th element of $\mathbf{b}$. It is easy to show that $\beta_j \gamma_j = 0$ for all $j \neq i$. When $w_{ij}$ satisfies $w_{ij} = w_{ii}$, according to Eq. (12) we have $\gamma_j = 0$ and further $\mathbf{w}_i \mathbf{a}_j + b_j = \alpha - \beta_j \leq \alpha$ according to Eq. (9). When $w_{ij} = -w_{ii}$, based on Eq. (11) we can get $\beta_j = 0$ and then $\mathbf{w}_i \mathbf{a}_j + b_j = \alpha + \gamma_j \geq \alpha$. For $w_{ij}$ between those two extremes (i.e., $-w_{ii} < w_{ij} < w_{ii}$), $\gamma_j = \beta_j = 0$ according to Eqs. (11) and (12), which implies that $\mathbf{w}_i \mathbf{a}_j + b_j = \alpha$. Moreover, Eq. (10) implies that $\mathbf{w}_i \mathbf{a}_i + b_i = \alpha + \sum_{k \neq i}(\beta_k + \gamma_k) \geq \alpha$. We define sets as $\mathcal{S}_1 = \{j | w_{ij} = w_{ii}, j \neq i\}$, $\mathcal{S}_2 = \{j | -w_{ii} < w_{ij} < w_{ii}\}$, $\mathcal{S}_3 = \{j | w_{ij} = -w_{ii}\}$, and $\mathcal{S}_4 = \{i\}$. Then a feasible $\mathbf{w}_i$ is a stationary point of problem (8) if and only if $\max_{j \in \mathcal{S}_1 \cup \mathcal{S}_2}\{\mathbf{w}_i \mathbf{a}_j + b_j\} \leq \min_{k \in \mathcal{S}_2 \cup \mathcal{S}_3 \cup \mathcal{S}_4}\{\mathbf{w}_i \mathbf{a}_k + b_k\}$. If there exist a pair of indices $(j, k)$, where $j \in \mathcal{S}_1 \cup \mathcal{S}_2$ and $k \in \mathcal{S}_2 \cup \mathcal{S}_3 \cup \mathcal{S}_4$, satisfying $\mathbf{w}_i \mathbf{a}_j + b_j > \mathbf{w}_i \mathbf{a}_k + b_k$, $\{j, k\}$ is called a violating pair. If the current estimation $\mathbf{w}_i$ is not an optimal solution, there should exist some violating pairs. Our SMO algorithm updates a violating pair at one step by choosing the most violating pair $\{j, k\}$ with $j$ and $k$ defined as $j = \arg \max_{l \in \mathcal{S}_1 \cup \mathcal{S}_2}\{\mathbf{w}_i \mathbf{a}_l + b_l\}$ and $k = \arg \min_{l \in \mathcal{S}_2 \cup \mathcal{S}_3 \cup \mathcal{S}_4}\{\mathbf{w}_i \mathbf{a}_l + b_l\}$. We define the update rule for $w_{ij}$ and $w_{ik}$ as $\tilde{w}_{ij} = w_{ij} + t$ and $\tilde{w}_{ik} = w_{ik} - t$. By noting that $j$ cannot be $i$, $t$ should satisfy the following constraints to make the updated solution feasible:

$$\text{when } k = i, \ t - w_{ik} \leq w_{ij} + t \leq w_{ik} - t, \ t - w_{ik} \leq w_{il} \leq w_{ik} - t \ \forall l \neq j \& l \neq k$$
$$\text{when } k \neq i, \ -w_{ii} \leq w_{ij} + t \leq w_{ii}, \ -w_{ii} \leq w_{ik} - t \leq w_{ii}.$$

When $k = i$, there will be a constraint on $t$ as $t \leq e \equiv \min\left(\frac{w_{ik} - w_{ij}}{2}, \min_{l \neq j \& l \neq k}(w_{ik} - |w_{il}|)\right)$ and otherwise $t$ will satisfy $c \leq t \leq e$ where $c = \max(w_{ik} - w_{ii}, -w_{ij} - w_{ii})$ and $e = \min(w_{ii} - w_{ij}, w_{ii} + w_{ik})$. Then the optimization problem for $t$ can be unified as

$$\min_t \frac{a_{jj} + a_{ii} - 2a_{ji}}{2} t^2 + (\mathbf{w}_i \mathbf{a}_j + b_j - \mathbf{w}_i \mathbf{a}_i - b_i)t \quad \text{s.t. } c \leq t \leq e,$$

where for the case that $k = i$, $c$ is set to be $-\infty$. This problem has an analytical solution as $t = \min\left(e, \max\left(c, \frac{\mathbf{w}_i \mathbf{a}_i + b_i - \mathbf{w}_i \mathbf{a}_j - b_j}{a_{jj} + a_{ii} - 2a_{ji}}\right)\right)$. We update each row of $\mathbf{W}$ one by one until convergence.

# 4 Experiments

In this section, we test the empirical performance of our proposed methods in some toy data and real-world problems.

## 4.1 Toy Problems

We first use one UCI dataset, i.e., diabetes data, to analyze the learned $\mathbf{W}$ matrix. The diabetes data consist of 768 data points from two classes. We randomly select $p$ percent of data points to form the training set of two learning tasks respectively. The regularization parameters $\lambda_1$ and $\lambda_2$ are fixed as 1 and the number of nearest neighbors is set to 5. When $p = 20$ and $p = 40$, the means of the estimated $\mathbf{W}$ over 10 trials are $\begin{bmatrix} 0.1025 & 0.1011 \\ 0.0980 & 0.1056 \end{bmatrix}$ and $\begin{bmatrix} 0.1014 & 0.1004 \\ 0.1010 & 0.1010 \end{bmatrix}$. This result shows $w_{ij}$ ($j \neq i$) is very close to $w_{ii}$ for $i = 1, 2$. This observation implies our method can find that these two tasks are positive correlated which matches our expectation since those two tasks are from the same distribution.

For the second experiment, we randomly select $p$ percent of data points to form the training set of two learning tasks respectively but differently we flip the labels of one task so that those two tasks should be negatively correlated. The matrices $\mathbf{W}$'s learned for $p = 20$ and $p = 40$ are $\begin{bmatrix} 0.1019 & -0.1017 \\ -0.1007 & 0.1012 \end{bmatrix}$ and $\begin{bmatrix} 0.1019 & -0.0999 \\ -0.0997 & 0.1038 \end{bmatrix}$. We can see that $w_{ij}$ ($j \neq i$) is very close to $-w_{ii}$ for $i = 1, 2$, which is what we expect.

As the third problem, we construct two learning tasks as in the first one but flip 50% percent of the class labels in each class of those two tasks. Here those two tasks can be viewed as unrelated tasks since the label assignment is random. The estimated matrices $\mathbf{W}$'s for $p = 20$ and $p = 40$ are $\begin{bmatrix} 0.1575 & 0.0144 \\ 0.0398 & 0.1281 \end{bmatrix}$ and $\begin{bmatrix} 0.1015 & -0.0003 \\ 0.0081 & 0.1077 \end{bmatrix}$, where $w_{ij}$ ($i \neq j$) is much smaller than $w_{ii}$. From the structure of the estimations, we can see that those two tasks are more likely to be unrelated, matching our expectation. In summary, our method can learn the positive correlations, negative correlations and task unrelatedness for those toy problems.

## 4.2 Experiments on Classification Problems

Two multi-task classification problems are used in our experiments. The first problem we investigate is a handwritten letter classification application consisting of seven tasks each of which is to distinguish two letters. The corresponding letters for each task to classify are: c/e, g/y, m/n, a/g, a/o, f/t and h/n. Each class in each task has about 1000 data points which have 128 features corresponding to the pixel values of hand-

Table 2: Comparison of classification errors of different methods on the two classification problems in the form of mean±std.

|  | Letter | USPS |
|---|---|---|
| KNN | 0.0775±0.0053 | 0.0445±0.0131 |
| mtLMNN | 0.0511±0.0053 | 0.0141±0.0038 |
| MTFL | 0.0505±0.0038 | 0.0140±0.0025 |
| MT-KNN(hinge) | **0.0466±0.0023** | **0.0114±0.0013** |
| MT-KNN(square) | 0.0494±0.0028 | 0.0124±0.0014 |

written letter images. The second one is the USPS digit classification problem and it consists of nine binary classification tasks each of which is to classify two digits. Each task contains about 1000 data points with 255 features for each class.

Here the similarity function we use is a heat kernel $s(i,j) = \exp\{-\frac{\|\mathbf{x}_i - \mathbf{x}_j\|_2^2}{2\sigma^2}\}$ where $\sigma$ is set to the mean pairwise Euclidean distance among training data. We use 5-fold cross validation to determine the optimal $\lambda_1$ and $\lambda_2$ whose candidate values are chosen from $n \times \{0.01, 0.1, 0.5, 1, 5, 10, 100\}$ and the optimal number of nearest neighbors from $\{5, 10, 15, 20\}$. The classification error is used as the performance measure. We compare our method, which is denoted as MT-KNN, with the KNN classifier which is a single-task learning method, the multi-task large margin nearest neighbor (mtLMNN) method [14][1] which is a multi-task local learning method based on the homogeneous neighborhood, and the multi-task feature learning (MTFL) method [2] which is a global method for multi-task learning. By uti-

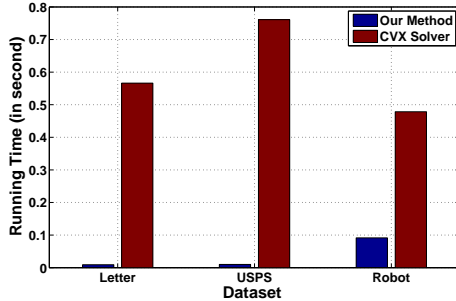

Figure 2: Comparison on average running time over 100 trials between our proposed coordinate descent methods and the CVX solver on classification and regression problems.

lizing hinge and square losses, we also consider two variants of our MT-KNN method. To mimic the real-world situation where the training data are usually limited, we randomly select 20% of the whole data as training data and the rest to form the test set. The random selection is repeated for 10 times and we record the results in Table 2. From the results, we can see that our method MT-KNN is better than KNN, mtLMNN and MTFL methods, which demonstrates that the introduction of the heterogeneous neighborhood is helpful to improve the performance. For different loss functions utilized by our method, MT-KNN with hinge loss is better than that with square loss due to the robustness of the hinge loss against the square loss.

For those two problems, we also compare our proposed coordinate descent method described in Table 1 with some off-the-shelf solvers such as the CVX solver [11] with respect to the running time. The platform to run the experiments is a desktop with Intel i7 CPU 2.7GHz and 8GB RAM and we use Matlab 2009b for implementation and experiments. We record the average running time over 100 trials in Figure 2 and from the results we can see that on the classification problems above, our proposed coordinate descent method is much faster than the CVX solver which demonstrates the efficiency of our proposed method.

## 4.3 Experiments on Regression Problems

Here we study a multi-task regression problem to learn the inverse dynamics of a seven degree-of-freedom SARCOS anthropomorphic robot arm.[2] The objective is to predict seven joint torques based

on 21 input features, corresponding to seven joint positions, seven joint velocities and seven joint accelerations. So each task corresponds to the prediction of one torque and can be formulated as a regression problem. Each task has 2000 data points. The similarity function used here is also the heat kernel and 5-fold cross validation is used to determine the hyperparameters, i.e., $\lambda_1$, $\lambda_2$ and $k$. The performance measure used is normalized mean squared error (nMSE), which is mean squared error on the test data divided by the variance of the ground truth. We compare our method denoted by MT-KR with single-task kernel regression (KR), the multi-task feature learning (MTFL) under different configurations on the size of the training set. Compared with KR and MTFL methods, our method achieves better performance over different sizes of the training sets. Moreover, for our proposed coordinate descent method introduced in section 3.1, we compare it with CVX solver and record the results in the last two columns of Figure 2. We find the running time of our proposed method is much smaller than that of the CVX solver which demonstrates that the proposed coordinate descent method can speed up the computation of our MT-KR method.

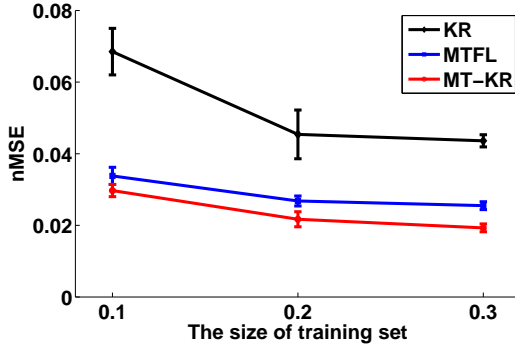

Figure 3: Comparison of different methods on the robot arm application when varying the size of the training set.

## 4.4 Sensitivity Analysis

Here we test the sensitivity of the performance with respect to the number of nearest neighbors. By changing the number of nearest neighbors from 5 to 40 at an interval of 5, we record the mean of the performance of our method over 10 trials in Figure 4. From the results, we can see our method is not very sensitive to the number of nearest neighbors, which makes the setting of $k$ not very difficult.

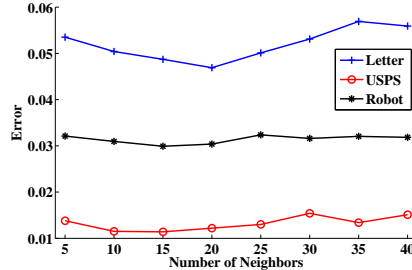

Figure 4: Sensitivity analysis of the performance of our method with respect to the number of nearest neighbors at different data sets.

## 5 Conclusion

In this paper, we develop local learning methods for multi-task classification and regression problems. Based on an assumption that all task pairs contributes to each other almost equally, we propose regularized objective functions and develop efficient coordinate descent methods to solve them. Up to here, each task in our studies is a binary classification problem. In some applications, there may be more than two classes in each task. So we are interested in an extension of our method to multi-task multi-class problems. Currently the task-specific weights are shared by all data points from one task. One interesting research direction is to investigate a localized variant where different data points have different task-specific weights based on their locality structure.

## Acknowledgment

Yu Zhang is supported by HKBU 'Start Up Grant for New Academics'.

## Footnotes

[1] http://www.cse.wustl.edu/~kilian/code/files/mtLMNN.zip

[2] http://www.gaussianprocess.org/gpml/data/

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
