[Reviews · NeurIPS 2013]

Submitted by Assigned_Reviewer_4

Summary: This paper presents a multitask learning method which entails jointly solving a collection of k-nearest neighbor (kNN) based prediction tasks, leveraging the relationships among the tasks. Whereas single task learning for kNN would only consider neighbors from the task which the test point belongs to (referred to as "homogeneous neighborhood" in the paper), the multitask variant proposed here considers neighbors from all tasks (referred to as "heterogeneous neighborhood" in the paper), suitably weighting the contribution of each neighbor by the pairwise similarity between the task the test point belongs o and the task the neighbor belongs to. The pairwise task similarities are learned from data. Experimental results show that the proposed method performs better than a kNN based multitask learning method anda global multitask learning method that learns a common feature represent of all tasks and learns predictors using that representation.

Quality: The proposed model makes sense, especially the way a local learning problem (neighborhood based kNN) has been reformulated as a global learning problem (like SVM) and then cast as a standard global multitask learning problem.

Clarity: The paper is well-written and the idea is easy to follow. Experimental results are well presented.

Originality: The idea of learning and using pairwise similarities isn't totally novel (see comments below) but doing it in the framework of local learning is kind of still novel.

Significance: The paper may be of moderate significance, given a large-body of already existing work in multitask learning (some of which uses learned pairwise task similarities).

Paper Strengths:

- Despite a large amount of work on global multitask learning methods, multitask learning for local methods such as kNN is relatively less looked at. Therefore, the paper is novel from that perspective.

- The proposed method is simple and intuitive.

- The method can be applied for both classification and regression.

- The method can learn pairwise task relationships (although a number of other recent works have also done this but for global prediction models).

- An efficient optimization procedure is given for learning the model parameters.

- Experimental results are impressive.

Paper Weaknesses:

- The authors seem to be unaware of some relevant works on multitask local/global learning (please see below in comments), some of which should have been compared against experimentally.

- The comment on lines 68-69 (existing work is limited to multitask kNN classification methods) isn't really true. For instance, Caruana [8] proposed a kNN based multitask regression method based on learning common weight (across all tasks) for each feature to be used while computing pairwise distances between examples.

Comments:

- There is a bunch of methods that improve upon mtLMNN (one of the classification baselines used in the paper):
(1) "Geometry preserving multi-task metric learning" (Yang et al, 2012)
(2) "Multi-Task Low-Rank Metric Learning Based on Common Subspace" (Yang et al, 2011)
A comparison with some of these should have been done.

- There should be a comparison with some global multitask learning baseline that learns pairwise task relationships just like the proposed method. I suggest the authors to compare with the following method:
"A convex formulation for learning task relationships in multi-task learning" (Zhang and Yeung, 2010).

- It would be good to have an experiment where the number of training samples per task is varied from, say, 5% to 20% or 30% (maybe with increments of 5%) and see how the different methods behave.

- It is not clear how the method will perform in high dimensions where pairwise distances computing in the original feature space may be unreliable. Please comment.
Summary: It's a good but sort of a borderline paper. The idea is simple and intuitive. A more thorough experimental evaluation (some baselines suggested in the review) would make it a stronger paper.

Submitted by Assigned_Reviewer_7

This paper proposes a local learning method for multiple task learning that generalizes kNN classifier by combining data points from multiple tasks. The authors introduce a weighted decision function by using task-specific weights.

The idea of casting multi-task local learning into an optimization problem is interesting. In this approach, a global weight w_{ij} is learnt for each pair of tasks: Tasks i and j. Since the base model is local learning method, it will be great if the authors can explore local weight scheme in the future. w_{ij} is set for all the feature space, it may not be able to capture the difference among different regions.

For a local learning method, how to set a proper similarity function is extremely important. In this paper, the authors use some predefined similarity function. They might want to discuss how the proposed approach works if different similarity functions are applied. If the algorithm works particularly well on certain functions but not on others, it would be better to explain why this happens.

Some more discussions are needed to explain experimental results. In the 'letter' experiment, there are binary tasks as c/e, g/y, m/n, a/g, a/o, f/t and h/n. It seems that there are almost no or very low correlations between any two of these tasks. In this case, w_{ij} should be small, and the multi-task problem reduces to multiple single task problems. It is worth explaining why the proposed method has better performance than the other methods. Will it possibly be due to the use of better similarity functions? In the toy experiment on positive correlation, the generated tasks actually can be regarded as the same task because the data are all randomly sampled from the same data set. It would be better to modify the distribution a little bit to make the two tasks highly correlated but not the same.

The authors might also want to discuss the effect of skewness on the proposed approach. For example, if one task has a lot more data points than the other tasks, will the results be dominated by the task with lots of data points?
Summary: In this paper, an optimization framework is proposed to infer a weighted combination of local learners. The authors provide sound solutions to the optimization problem.

Submitted by Assigned_Reviewer_9

The paper presents a set of novel Multi-Task binary-classification and regression algorithms based on a weighted K-NN prediction rule. The main idea is to make use of the labels of the nearest neighbors from all the tasks, in order to predict the score for a target sample (instead of using only the labels of the neighbors from the target task). The prediction provided by each of the nearest neighbors is weighted by a scalar which encodes the task-relationship between the task of the target sample and the task of the neighbor considered. The task-to-task weights are collected in a matrix and a global objective function is devised to minimize the prediction error on the training set. The objective function penalizes non-symmetric matrices and includes constraints to force the magnitude of the task-to-task weights to be smaller whenever the source and the target tasks differ. The penalties used for the classification tasks include the hinge-loss and the square-loss. For regression tasks the authors employ the square loss and replace the K-NN classification rule with a kernel-regression rule.
In all the settings the optimization is performed using a (block) coordinate-descent approach, where each sub-problem is solved using efficient ad-hoc algorithms. For classification, the training time complexity is reported to be linear in the number of tasks and log-linear (linearithmic) in the number of training samples from each task, however this complexity does not include the time for the K-NN search for each sample. Moreover, the training complexity of the regression algorithm does not seem to be provided.

The algorithm is tested on two classification tasks and one regression task and is compared with two other classical multi-task learning approaches. The results show that the proposed methods compare favorably w.r.t. the selected baselines, while the proposed training procedures result in a significant speedup w.r.t. the use of standard solvers (CVX).

The main novelty of the approach is the use of the nearest neighbors from all tasks, combined with the learned task-to-task relatedness matrix. For classification tasks this results in the possibility of learning positive, neutral and negative correlations between tasks, which in turn result in positive, zero, or negative weights assigned to each neighbor label. For regression task, the weighting allows to learn a linear relationship between the different tasks.

A concern is about the convergence of the coordinate-descent approach when used to minimize non-smooth functions (e.g. hinge loss). The authors claim that the algorithm: "can converge to a global optimum" since "all the subproblems are strictly convex", citing the results in [1]. However, the hypothesis for local convergence in [1] require the objective function to be smooth (C^2), while for global convergence the solutions in the optimization procedure are required to lie in a compact set.
A more in-depth discussion and explanation seems thus to be necessary.

[1] Bezdek, James C., and Richard J. Hathaway. "Convergence of alternating optimization." Neural, Parallel & Scientific Computations 11.4 (2003): 351-368.

The experimental evaluation seems rather limited:
- the classification datasets considered (USPS, Letter) do not seem to match the datasets used in the papers of the considered baselines
- the baseline selected only include multi-task methods which assumes that all the tasks are related, while more recent multi-task approaches (e.g. [2] and [3]) are moving away from this assumption; since the proposed method is also able to learn how the tasks are related to each other, a comparison should be made with such methods
- while using nearest neighbors from all the tasks seem to improve the performances (w.r.t using only the K-NNs of the target task), it is also supposed to increase the K-NN search time; a comparison of training-complexity or training-times w.r.t. the baseline methods would be appreciated.

The exposition is not always clear. Examples:

- in lines 40-42 the authors claim that "When the number of labeled data is not very large, the performance of the local learning methods is limited due to sparse local density". This is an important point for the proposed methodology, as it is one of the motivations for using the samples from all the tasks. It should thus definitely discussed better, or a citation should be provided.
- in lines 53-55 the authors write about "low-confidence" and high-confidence estimations. However, the notion of confidence of the estimations is never formalized for the methods in question, so that the claims result to be quite vague
- in lines 61-64 the authors write: "Since multi-task learning usually considers that the contribution of one task to another one equals that in the reverse direction, we define a regularizer to enforce the task-specific weight matrix to approach a symmetric matrix"; the requirement of symmetry in the task-to-task relationships is also an important point of the proposed approach and should thus be discussed more thoroughly, or a citation should be provided
- in lines 358-361 of the experimental section two baseline algorithm (mtLMNN and MTFL) are apparently introduced without any citation; while it is possible to infer the related publications from the acronyms the citations should be reported
- the URL reported in line 377 for the Letter dataset (http://multitask.cs.berkeley.edu/) does not seem to exist anymore

Minor comments
While the classification function is local, the task-to-task relationship matrix W is global and the learning of W is done in a global-fashion, rather than in a local, as it figures in the title.
Summary: The proposed multi-task approach is interesting and the authors have made a significant effort to devise efficient optimization procedures to solve the proposed problems. Additional efforts should be made to clarify the points related to the convergence with the hinge-loss, the training complexity including the K-NN search time and the necessity for simmetry in the task-to-task relationship matrix. Also, the choice of datasets and baselines should be better motivated.
Author Feedback

Author rebuttal: To all reviewers:

Thanks for your comments!

To Reviewer1:

Q: There is a bunch of methods that improve upon mtLMNN ...
A: Thank you for pointing out some relevant references! We will add discussion on them in the revision.

Q: The comment on lines 68-69 isn't really true ...
A: As you said, [8] has proposed a kNN-based multi-task regression model. We will revise line 68-69.

Q: There should be a comparison with ... baseline ... (Zhang and Yeung, 2010)
A: In experiments, we compare with the multi-task feature learning (MTFL), a global multi-task learning method. Even though MTFL has a different formulation with the method in (Zhang and Yeung, 2010), they both optimize with the trace norm regularization and hence have similar behaviors. So we just choose MTFL as a baseline.

Q: It would be good ... number of training samples per task is varied ... behave
A: We did experiments on varying size of the training set in the regression problem. We will add more experiments in the revision.

Q: It is not clear ... in high dimensions where pairwise distances ... unreliable ...
A: When handling high-dimensional data, pairwise distances may have small difference and cannot reflect the true similarity. In this case, we can first find some linear or nonlinear subspace which preserves some characteristics of the original space and then we can conduct our proposed method in the reduced space.

To Reviewer2:

Q: This global weight scheme ... reasonable because the base model ... capture the difference among different regions.
A: Inspired by [6,20], we assume that different regions of one pair of tasks share a global task relation reflected in w_{ij}. One future direction of our work, as discussed in the conclusion, is just to learn local task relations for different regions in each pair of tasks.

Q: They might want to discuss ... if different similarity functions …
A: We have tried other similarity functions (e.g., dot-product similarity or cosine similarity) on the datasets used in the experiments. However, the performance is not very good. We will add some discussion on this point.

Q: In the 'letter' experiment, there are binary ... worth explaining why ... has better performance …
A: In the 'letter' experiment, since some pairs of tasks have one class respectively to represent the same letter, e.g., the 3rd and 7th task sharing letter 'n' and the 4th and 5th task sharing letter 'a', we think there exist some pairs of tasks having large correlations. This dataset is used by one research group in Stanford University as one benchmark dataset for multi-task learning and we just download from their website. Moreover, the 'letter' dataset has been used in some existing multi-task learning works (e.g., 'Transfer Metric Learning by Learning Task Relationships', KDD2010; 'Multi-Task Boosting by Exploiting Task Relationships', ECML2012) and the experiments in those papers have demonstrated that the 'letter' dataset can be used as one benchmark dataset for multi-task learning.

Q: In the toy experiment ... It may not be a proper testbed ...
A: Since we need to know the ground truth of the task correlations for the toy problem, we generate two tasks by sampling from one dataset in which we have an idea of the true task correlations. And then based on the ground truth, we can test how well our method works on this toy problem. Based on the consideration, we design the toy problem like that.

Q: if one task has a lot more data points ..., will the results be dominated ...
A: When there is some skewness, we can use the reciprocal of the number of the training data points in each task as a weight for the loss of that task, which can eliminate the effect of skewness to some extent. In our experiments, different tasks have the same number of training data points and hence there is no need to do that.

To Reviewer3:

Q: A concern is about the convergence ... to be necessary.
A: For the convergence guarantee of coordinate descent methods for nonsmooth problems, the work ‘Convergence of a block coordinate descent method for nondifferentiable minimization’ proves that the requirement is just that each subproblem has a unique minimizer. Our problem satisfies this condition and hence can converge.

Q: the classification datasets ... not ... match ... considered baselines
A: Those datasets were downloaded from a website about multi-task learning from Stanford University. Moreover, those datasets have been used in some existing multi-task learning works, e.g., 'Transfer Metric Learning by Learning Task Relationships' published in KDD2010 and 'Multi-Task Boosting by Exploiting Task Relationships' published in ECML2012.

Q: the baseline selected ... a comparison ... made ... with methods
A: Actually the MTFL method we compared is the approach in [2].

Q: while using nearest neighbor ...; a comparison of training-complexity ... be appreciated
A: In our experiments, we use K-D tree and hashing to accelerate K-NN search. We will add comparison on the training times with respect to the baseline.

Q: in lines 40-42 ... a citation ... provided
A: We will add some references (e.g., [17,8,14]) and some discussion on that point.

Q: in lines 53-55 ... confidence formalized ...
A: We will define ‘confidence’ formally in the revision.

Q: in lines 61-64 ... a citation should be provided
A: We will add some references (e.g., [6,20]) and some discussion on that point.

Q: in lines 358-361 ... the citations should be reported
A: We will add citations to the two baseline algorithms in the revision.

Q: the URL reported in line 377 ... not ... exist
A: The website has been closed. We will put the dataset on our homepage.

Q: While the classification function is local, the ... W is global ... in the title
A: The 'local learning algorithm' in the title refers to the classification function but not the task relations. We will make it clearer.